# RETRIEVAL-AUGMENTED GENERATION FOR CODE SUMMARIZATION VIA HYBRID GNN

**Shangqing Liu[1]\*, Yu Chen[2]†, Xiaofei Xie[1]†, Jingkai Siow[1], Yang Liu[1]**
[1] **Nanyang Technology University**
[2] **Rensselaer Polytechnic Institute**

## ABSTRACT

Source code summarization aims to generate natural language summaries from structured code snippets for better understanding code functionalities. However, automatic code summarization is challenging due to the complexity of the source code and the language gap between the source code and natural language summaries. Most previous approaches either rely on retrieval-based (which can take advantage of similar examples seen from the retrieval database, but have low generalization performance) or generation-based methods (which have better generalization performance, but cannot take advantage of similar examples). This paper proposes a novel retrieval-augmented mechanism to combine the benefits of both worlds. Furthermore, to mitigate the limitation of Graph Neural Networks (GNNs) on capturing global graph structure information of source code, we propose a novel attention-based dynamic graph to complement the static graph representation of the source code, and design a hybrid message passing GNN for capturing both the local and global structural information. To evaluate the proposed approach, we release a new challenging benchmark, crawled from diversified large-scale open-source *C* projects (total **95k+** unique functions in the dataset). Our method achieves the state-of-the-art performance, improving existing methods by **1.42**, **2.44** and **1.29** in terms of BLEU-4, ROUGE-L and METEOR.

## 1   INTRODUCTION

With software growing in size and complexity, developers tend to spend nearly 90% (Wan et al., 2018) effort on software maintenance (*e.g.*, version iteration and bug fix) in the completed life cycle of software development. Source code summary, in the form of natural language, plays a critical role in the comprehension and maintenance process and greatly reduces the effort of reading and comprehending programs. However, manually writing code summaries is tedious and time-consuming, and with the acceleration of software iteration, it has become a heavy burden for software developers. Hence, source code summarization which automates concise descriptions of programs is meaningful.

Automatic source code summarization is a crucial yet far from the settled problem. The key challenges include: 1) the source code and the natural language summary are heterogeneous, which means they may not share common lexical tokens, synonyms, or language structures and 2) the source code is complex with complicated logic and variable grammatical structure, making it hard to learn the semantics. Conventionally, information retrieval (IR) techniques have been widely used in code summarization (Eddy et al., 2013; Haiduc et al., 2010; Wong et al., 2015; 2013). Since code duplication (Kamiya et al., 2002; Li et al., 2006) is common in "big code" (Allamanis et al., 2018), early works summarize the new programs by retrieving the similar code snippet in the existing code database and use its summary directly. Essentially, the retrieval-based approaches transform the code summarization to the code similarity calculation task, which may achieve promising performance on similar programs, but are limited in generalization, *i.e.* they have poorer performance on programs that are very different from the code database.

---

\*Contact:shangqin001@e.ntu.edu.sg
†Corresponding authors

To improve the generalization performance, recent works focus on generation-based approaches. Some works explore Seq2Seq architectures (Bahdanau et al., 2014; Luong et al., 2015) to generate summaries from the given source code. The Seq2Seq-based approaches (Iyer et al., 2016; Hu et al., 2018a; Alon et al., 2018) usually treat the source code or abstract syntax tree parsed from the source code as a sequence and follow a paradigm of encoder-decoder with the attention mechanism for generating a summary. However, these works only rely on sequential models, which are struggling to capture the rich semantics of source code *e.g.*, control dependencies and data dependencies. In addition, generation-based approaches typically cannot take advantage of similar examples from the retrieval database, as retrieval-based approaches do.

To better learn the semantics of the source code, Allamanis et al. (Allamanis et al., 2017) lighted up this field by representing programs as graphs. Some follow-up works (Fernandes et al., 2018) attempted to encode more code structures (*e.g.*, control flow, program dependencies) into code graphs with graph neural networks (GNNs), and achieved the promising performance than the sequence-based approaches. Existing works (Allamanis et al., 2017; Fernandes et al., 2018) usually convert code into graph-structured input during preprocessing, and directly consume it via modern neural networks (*e.g.*, GNNs) for computing node and graph embeddings. However, most GNN-based encoders only allow message passing among nodes within a $k$-hop neighborhood (where $k$ is usually a small number such as 4) to avoid over-smoothing (Zhao & Akoglu, 2019; Chen et al., 2020a), thus capture only local neighborhood information and ignore global interactions among nodes. Even there are some works (Li et al., 2019) that try to address this challenging with deep GCNs (i.e., 56 layers) (Kipf & Welling, 2016) by the residual connection (He et al., 2016), however, the computation cost cannot endure in the program especially for a large and complex program.

To address these challenges, we propose a framework for automatic code summarization, namely Hybrid GNN *(HGNN)*. Specifically, from the source code, we first construct a code property graph (CPG) based on the abstract syntax tree (AST) with different types of edges (*i.e.*, Flow To, Reach). In order to combine the benefits of both retrieval-based and generation-based methods, we propose a *retrieval-based augmentation mechanism* to retrieve the source code that is most similar to the current program from the retrieval database (excluding the current program itself), and add the retrieved code as well as the corresponding summary as auxiliary information for training the model. In order to go beyond local graph neighborhood information, and capture global interactions in the program, we further propose an attention-based dynamic graph by learning global attention scores (*i.e.*, edge weights) in the augmented static CPG. Then, a hybrid message passing (HMP) is performed on both static and dynamic graphs. We also release a new code summarization benchmark by crawling data from popular and diversified projects containing **95k+** functions in *C* programming language and make it public [1]. We highlight our main contributions as follows:

- We propose a general-purpose framework for automatic code summarization, which combines the benefits of both retrieval-based and generation-based methods via a retrieval-based augmentation mechanism.

- We innovate a Hybrid GNN by fusing the static graph (based on code property graph) and dynamic graph (via structure-aware global attention mechanism) to mitigate the limitation of the GNN on capturing global graph information.

- We release a new challenging *C* benchmark for the task of source code summarization.

- We conduct an extensive experiment to evaluate our framework. The proposed approach achieves the state-of-the-art performance and improves existing approaches by **1.42**, **2.44** and **1.29** in terms of BLEU-4, ROUGE-L and METEOR metrics.

## 2 HYBRID GNN FRAMEWORK

In this section, we introduce the proposed framework Hybrid GNN *(HGNN)*, as shown in Figure 1, which mainly includes four components: 1) Retrieval-augmented Static Graph Construction *(c.f.,* Section 2.2), which incorporates retrieved code-summary pairs to augment the original code for learning. 2) Attention-based Dynamic Graph Construction *(c.f.,* Section 2.3), which allows message passing among any pair of nodes via a structure-aware global attention mechanism. 3) *HGNN*, *(c.f.,*

---

[1]https://github.com/shangqing-liu/CCSD-benchmark-for-code-summarization

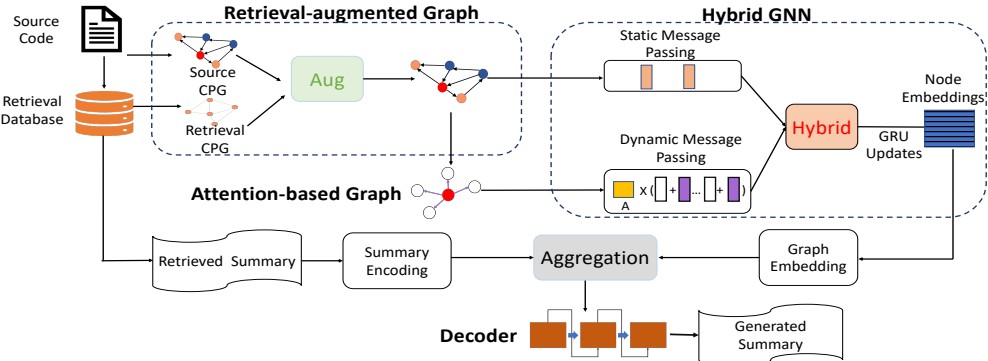

Figure 1: The overall architecture of the proposed *HGNN* framework.

Section 2.4), which incorporates information from both static graphs and dynamic graphs with Hybrid Message Passing. 4) Decoder (*c.f.,* Section 2.5), which utilizes an attention-based LSTM (Hochreiter & Schmidhuber, 1997) model to generate a summary.

## 2.1 PROBLEM FORMULATION

In this work, we focus on generating natural language summaries for the given functions (Wan et al., 2018; Zhang et al., 2020). A simple example is illustrated in Listing 1, which is crawled from Linux Kernel. Our goal is to generate the best summary "set the time of day clock" based on the given source code. Formally, we define a dataset as $D = \{(c,s)|c \in C, s \in S\}$, where $c$ is the source code of a function in the function set $C$ and $s$ represents its targeted summary in the summary set $S$. The task of code summarization is, given a source code $c$, to generate the best summary consisting of a sequence of tokens $\hat{s} = (t_1, t_2, ..., t_T)$ that maximizes the conditional likelihood $\hat{s} = \mathrm{argmax}_s P(s|c)$.

```
Source Code:
  int pdc_tod_set(unsigned long sec, unsigned long usec){
      int retval; unsigned long flags;
      spin_lock_irqsave(&pdc_lock, flags);
      retval = mem_pdc_call(PDC_TOD, PDC_TOD_WRITE, sec, usec);
      spin_unlock_irqrestore(&pdc_lock, flags);
      return retval;
  }
Ground-Truth: set the time of day clock
```

Listing 1: An example in our dataset crawled from Linux Kernel.

## 2.2 RETRIEVAL-AUGMENTED STATIC GRAPH

### 2.2.1 GRAPH INITIALIZATION

The source code of a function can be represented as Code Property Graph (CPG) (Yamaguchi et al., 2014), which is built on the abstract syntax tree (AST) with different type of edges (*i.e.*, Flow To, Control, Define/Use, Reach). Formally, one raw function $c$ could be represented by a multi-edged graph $g(\mathcal{V}, \mathcal{E})$, where $\mathcal{V}$ is the set of AST nodes, $(v, u) \in \mathcal{E}$ denotes the edge between the node $v$ and the node $u$. A node $v$ consists of two parts: the *node sequence* and the *node type*. An illustrative example is shown in Figure 2. For example, in the red node, $a\%2 == 0$ is the node sequence and $Condition$ is the node type. An edge $(v, u)$ has a type, named *edge type*, *e.g.*, AST type and Flow To type. For more details about the CPG, please refer to Appendix A.

**Initialization Representation.** Given a CPG, we utilize a BiLSTM to encode its nodes. We represent each token of the node sequence and each edge type using the learned embedding matrix $\boldsymbol{E}^{seqtoken}$ and $\boldsymbol{E}^{edgetype}$, respectively. Then nodes and edges of the CPG can be encoded as:

$$
\begin{aligned}
\boldsymbol{h}_1, ..., \boldsymbol{h}_l &= \mathrm{BiLSTM}(\boldsymbol{E}_{v,1}^{seqtoken}, ..., \boldsymbol{E}_{v,l}^{seqtoken}) \\
encode\_node(v) &= [\boldsymbol{h}_l^{\rightarrow}; \boldsymbol{h}_1^{\leftarrow}] \\
encode\_edge(v, u) &= \boldsymbol{E}_{v,u}^{edgetype} \ \ if \ \ (v, u) \in \mathcal{E} \ \ else \ \ \boldsymbol{0}
\end{aligned}
\tag{1}
$$

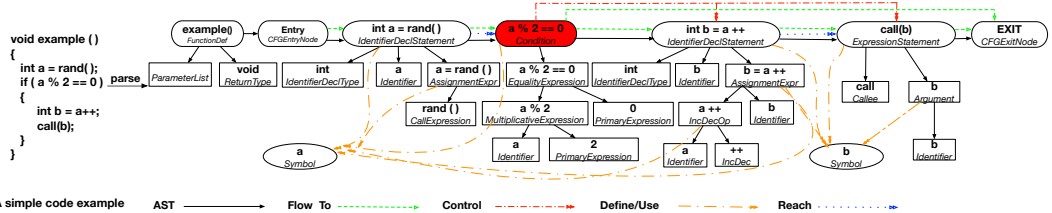

Figure 2: An example of Code Property Graph (CPG).

where $l$ is the number of tokens in the node sequence of $v$. For the sake of simplicity, in the following section, we use $\boldsymbol{h}_v$ and $\boldsymbol{e}_{v,u}$ to represent the embedding of the node $v$ and the edge $(v, u)$, respectively, *i.e.*, $encode\_node(v)$ and $encode\_edge(v, u)$. Given the source code $c$ of a function as well as the CPG $g(\mathcal{V}, \mathcal{E})$, $\boldsymbol{H}_c \in \mathbb{R}^{m \times d}$ denotes the initial node matrix of the CPG, where $m$ is the total number of nodes in the CPG and $d$ is the dimension of the node embedding.

### 2.2.2 RETRIEVAL-BASED AUGMENTATION

While retrieval-based methods can perform reasonably well on examples that are similar to those examples from a retrieval database, they typically have low generalization performance and might perform poorly on dissimilar examples. On the contrary, generation-based methods usually have better generalization performance, but cannot take advantage of similar examples from the retrieval database. In this work, we propose to combine the benefits of the two worlds, and design a retrieval-augmented generation framework for the task of code summarization.

In principle, the goal of code summarization is to learn a mapping from source code $c$ to the natural language summary $s = f(c)$. In other words, for any source code $c'$, a code summarization system can produce its summary $s' = f(c')$. Inspired by this observation, conceptually, we can derive the following formulation $s = f(c) - f(c') + s'$. This tells us that we can actually compute the semantic difference between $c$ and $c'$, and further obtain the desired summary $s$ for $c$ by considering both the above semantic difference and $s'$ which is the summary for $c'$. Mathmatically, our goal becomes to learn a function which takes as input $(c, c', s')$, and outputs the summary $s$ for $c$, that is, $s = g(c, c', s')$. This motivates us to design our Retrieval-based Augmentation mechanism, as detailed below.

**Step 1: Retrieving.** For each sample $(c, s) \in D$, we retrieve the most similar sample: $(c', s') = \operatorname{argmax}_{(c',s') \in D'} sim(c, c')$, where $c \neq c'$, $D'$ is a given retrieval database and $sim(c, c')$ is the text similarity. Following Zhang et al. (2020), we utilize Lucene for retrieval and calculate the similarity score $z$ between the source code $c$ and the retrieved code $c'$ via dynamic programming (Bellman, 1966), namely, $z = 1 - \frac{dis(c,c')}{max(|c|,|c'|)}$, where $dis(c, c')$ is the text edit distance.

**Step 2: Retrieved Code-based Augmentation.** Given the retrieved source code $c'$ for the current sample $c$, we adopt a fusion strategy to inject retrieved semantics into the current sample. The fusion strategy is based on their initial graph representations ($\boldsymbol{H}_c$ and $\boldsymbol{H}_{c'}$) with an attention mechanism:

- To capture the relevance between $c$ and $c'$, we design an attention function, which computes the attention score matrix $\boldsymbol{A}^{aug}$ based on the embeddings of each pair of nodes in CPGs of $c$ and $c'$:

$$\boldsymbol{A}^{aug} \propto \exp(\text{ReLU}(\boldsymbol{H}_c \boldsymbol{W}^C)\text{ReLU}(\boldsymbol{H}_{c'} \boldsymbol{W}^Q)^T) \quad (2)$$

where $\boldsymbol{W}^C, \boldsymbol{W}^Q \in \mathbb{R}^{d \times d}$ is the weight matrix with $d$-dim embedding size and ReLU is the rectified linear unit.

- We then multiply the attention matrix $\boldsymbol{A}^{aug}$ with the retrieved representation $\boldsymbol{H}_{c'}$ to inject the retrieved features into $\boldsymbol{H}_c$:

$$\boldsymbol{H}'_c = z \boldsymbol{A}^{aug} \boldsymbol{H}_{c'} \quad (3)$$

where $z \in [0, 1]$ is the similarity score and computed from Step 1, which is introduced to weaken the negative impact of $c'$ on the original training data $c$, *i.e.*, when the similarity of $c$ and $c'$ is low.

- Finally, we merge $\boldsymbol{H}'_c$ and the original $\boldsymbol{H}_c$ to get the final representation of $c$.

$$\boldsymbol{comp} = \boldsymbol{H}_c + \boldsymbol{H}'_c \quad (4)$$

where $\boldsymbol{comp}$ is the augmented node representation additionally encoding the retrieved semantics.

**Step 3: Retrieved Summary-based Augmentation.** We further encode the retrieved summary $s'$ with another BiLSTM model. We represent each token $t_i'$ of $s'$ using the learned embedding matrix $\boldsymbol{E}^{seqtoken}$. Then $s'$ can be encoded as:

$$\boldsymbol{h}_{t_1'}, ..., \boldsymbol{h}_{t_T'} = \text{BiLSTM}(\boldsymbol{E}_{t_1'}^{seqtoken}, ..., \boldsymbol{E}_{t_T'}^{seqtoken}) \tag{5}$$

where $\boldsymbol{h}_{t_i'}$ is the hidden state of the BiLSTM model for the token $t_i'$ in $s'$ and $T$ is the length of $s'$. We multiply $[\boldsymbol{h}_{t_1'}; ...; \boldsymbol{h}_{t_T'}]$ with the similarity score $z$, computed from Step 1, and concatenate it with the graph encoding results (i.e., the GNN encoder outputs) to obtain the input, namely, $[\text{GNN}_{\text{output}}; z\boldsymbol{h}_{t_1'}; ...; z\boldsymbol{h}_{t_T'}]$, to the decoder.

## 2.3 ATTENTION-BASED DYNAMIC GRAPH

Due to that GNN-based encoders usually consider the $k$-hop neighborhood, the global relation among nodes in the static graph (see Section 2.2.1) may be ignored. In order to better capture the global semantics of source code, based on the static graph, we propose to dynamically construct a graph via structure-aware global attention mechanism, which allows message passing among any pair of nodes. The attention-based dynamic graph can better capture the global dependency among nodes, and thus supplement the static graph.

**Structure-aware Global Attention.** The construction of the dynamic graph is motivated by the structure-aware self-attention mechanism proposed in Zhu et al. (2019). Given the static graph, we compute a corresponding dense adjacency matrix $\boldsymbol{A}^{dyn}$ based on a structure-aware global attention mechanism, and obtain the constructed graph, namely, *attention-based dynamic graph*.

$$\boldsymbol{A}_{v,u}^{dyn} = \frac{\text{ReLU}(\boldsymbol{h}_v^T \boldsymbol{W}^Q)(\text{ReLU}(\boldsymbol{h}_u^T \boldsymbol{W}^K) + \text{ReLU}(\boldsymbol{e}_{v,u}^T \boldsymbol{W}^R))^T}{\sqrt{d}} \tag{6}$$

where $\boldsymbol{h}_v, \boldsymbol{h}_u \in \boldsymbol{comp}$ are the augmented node embedding for any node pair $(v, u)$ in the CPG. Note that the global attention considers each pair of nodes of the CPG, regardless of whether there is an edge between them. $\boldsymbol{e}_{v,u} \in \mathbb{R}^{d_e}$ is the edge embedding and $\boldsymbol{W}^Q, \boldsymbol{W}^K \in \mathbb{R}^{d \times d}$, $\boldsymbol{W}^R \in \mathbb{R}^{d_e \times d}$ are parameter matrices, $d_e$ and $d$ are the dimensions of edge embedding and node embedding, respectively. The adjacency matrix $\boldsymbol{A}^{dyn}$ will be further row normalized to obtain $\tilde{\boldsymbol{A}}^{dyn}$, which will be used to compute dynamic message passing (see Section 2.4).

$$\tilde{\boldsymbol{A}}^{dyn} = \text{softmax}(\boldsymbol{A}^{dyn}) \tag{7}$$

## 2.4 HYBRID GNN

To better incorporate the information of the static graph and the dynamic graph, we propose the Hybrid Message Passing (HMP), which are performed on both retrieval-augmented static graph and attention-based dynamic graph.

**Static Message Passing.** For every node $v$ at each computation hop $k$ in the static graph, we apply an aggregation function to calculate the aggregated vector $\boldsymbol{h}_v^k$ by considering a set of neighboring node embeddings computed from the previous hop.

$$\boldsymbol{h}_v^k = \text{SUM}(\{\boldsymbol{h}_u^{k-1} | \forall u \in \mathcal{N}_{(v)}\}) \tag{8}$$

where $\mathcal{N}_{(v)}$ is a set of the neighboring nodes which are directly connected with $v$. For each node $v$, $\boldsymbol{h}_v^0$ is the initial augmented node embedding of $v$, *i.e.*, $\boldsymbol{h}_v \in \boldsymbol{comp}$.

**Dynamic Message Passing.** The node information and edge information are propagated on the attention-based dynamic graph with the adjacency matrices $\tilde{\boldsymbol{A}}^{dyn}$, defined as

$$\boldsymbol{h}_v^{'k} = \sum_u \tilde{\boldsymbol{A}}_{v,u}^{dyn}(\boldsymbol{W}^V \boldsymbol{h}_u^{'k-1} + \boldsymbol{W}^F \boldsymbol{e}_{v,u}) \tag{9}$$

where $v$ and $u$ are any pair of nodes, $\boldsymbol{W}^V \in \mathbb{R}^{d \times d}$, $\boldsymbol{W}^F \in \mathbb{R}^{d \times d_e}$ are learned matrices, and $\boldsymbol{e}_{v,u}$ is the embedding of the edge connecting $v$ and $u$. Similarly, $\boldsymbol{h}_v^{'0}$ is the initial augmented node embedding of $v$ in $\boldsymbol{comp}$.

**Hybrid Message Passing.** Given the static/dynamic aggregated vectors $\boldsymbol{h}_v^k / \boldsymbol{h}_v^{'k}$ for static and dynamic graphs, we fuse both vectors and feed the resulting vector to a Gated Recurrent Unit (GRU) to update node representations.

$$\boldsymbol{f}_v^k = \text{GRU}(\boldsymbol{f}_v^{k-1}, \text{Fuse}(\boldsymbol{h}_v^k, \boldsymbol{h}_v^{'k})) \tag{10}$$

where $\boldsymbol{f}_v^0$ is the augmented node initialization in $\boldsymbol{comp}$. The fusion function Fuse is designed as a gated sum of two inputs.

$$\text{Fuse}(\boldsymbol{a}, \boldsymbol{b}) = \boldsymbol{z} \odot \boldsymbol{a} + (1 - \boldsymbol{z}) \odot \boldsymbol{b} \quad \boldsymbol{z} = \sigma(\boldsymbol{W}_z[\boldsymbol{a}; \boldsymbol{b}; \boldsymbol{a} \odot \boldsymbol{b}; \boldsymbol{a} - \boldsymbol{b}] + \boldsymbol{b}_z) \tag{11}$$

where $\boldsymbol{W}_z$ and $\boldsymbol{b}_z$ are learnable weight matrix and vector, $\odot$ is the component-wise multiplication, $\sigma$ is a sigmoid function and $\boldsymbol{z}$ is a gating vector. After $n$ hops of GNN computation, we obtain the final node representation $\boldsymbol{f}_v^n$ and then apply max-pooling over all nodes $\{\boldsymbol{f}_v^n | \forall v \in \mathcal{V}\}$ to get the graph representation.

## 2.5 Decoder

The decoder is similar with other state-of-the-art Seq2seq models (Bahdanau et al., 2014; Luong et al., 2015) where an attention-based LSTM decoder is used. The decoder takes the input of the concatenation of the node representation and the representation of the retrieved summary $s'$, namely, $[\boldsymbol{f}_{v_1}^n; ...; \boldsymbol{f}_{v_m}^n; z\boldsymbol{h}_{t_1'}; ...; z\boldsymbol{h}_{t_T'}]$, where $m$ is the number of nodes in the input CPG graph. The initial hidden state of the decoder is the fusion (Eq. 11) of the graph representation and the weighted (i.e., multiply similarity score $z$) final state of the retrieved summary BiLSTM encoder.

We train the model with the cross-entropy loss, defined as $\mathcal{L} = \sum_t -\log P(s_t^* | c, s_{<t}^*)$, where $s_t^*$ is the word at the $t$-th position of the ground-truth output and $c$ is the source code of the function. To alleviate the exposure bias, we utilize schedule teacher forcing (Bengio et al., 2015). During the inference, we use beam search to generate final results.

## 3 Experiments

### 3.1 Setup

We evaluate our proposed framework against a number of state-of-the-art methods. Specifically, we classify the selected baseline methods into three groups: 1) Retrieval-based approaches: TF-IDF (Haiduc et al., 2010) and NNGen (Liu et al., 2018), 2) Sequence-based approaches: CODE-NN (Iyer et al., 2016; Barone & Sennrich, 2017), Transformer (Ahmad et al., 2020), Hybrid-DRL (Wan et al., 2018), Rencos (Zhang et al., 2020) and Dual model (Wei et al., 2019), 3) Graph-based approaches: SeqGNN (Fernandes et al., 2018). In addition, we implemented two another graph-based baselines: GCN2Seq and GAT2Seq, which respectively adopt the Graph Convolution (Kipf & Welling, 2016) and Graph Attention (Velickovic et al., 2018) as the encoder and a LSTM as the decoder for generating summaries. Note that Rencos (Zhang et al., 2020) combines the retrieval information into Seq2Seq model, we classify it into Sequence-based approaches. More detailed description about baselines and the configuration of *HGNN* can be found in the Appendix B and C.

Existing benchmarks (Barone & Sennrich, 2017; Hu et al., 2018b) are all based on high-level programming language *i.e.*, Java, Python. Furthermore, they have been confirmed to have extensive duplication, making the model overfit to the training data that overlapped with the testset (Fernandes et al., 2018; Allamanis, 2019). We are the first to explore neural summarization on *C* programming language, and make our *C* Code Summarization Dataset (CCSD) public to benefit academia and industry. We crawled from popular *C* repositories on GitHub and extracted function-summary pairs based on the documents of functions. After a deduplication process, we kept **95k+** unique function-summary pairs. To further test the model generalization ability, we construct in-domain functions and out-of-domain functions by dividing the projects into two sets, denoted as $a$ and $b$. For each project in $a$, we randomly select some of the functions in this project as the training data and the unselected functions are the in-domain validation/test data. All functions in projects $b$ are regarded as out-of-domain test data. Finally, we obtain 84,316 training functions, 4,432 in-domain validation functions, 4,203 in-domain test functions and 2,330 out-of-domain test functions. For the retrieval augmentation, we use the training set as the retrieval database, *i.e.*, $D' = D$ (see Step 1 in Section 2.2.2). For more details about data processing, please refer to Appendix D.

Table 1: Automatic evaluation results (in %) on the CCSD test set.

| Methods | In-domain | | | Out-of-domain | | | Overall | | |
|---|---|---|---|---|---|---|---|---|---|
| | BLEU-4 | ROUGE-L | METEOR | BLEU-4 | ROUGE-L | METEOR | BLEU-4 | ROUGE-L | METEOR |
| TF-IDF | 15.20 | 27.98 | 13.74 | 5.50 | 15.37 | 6.84 | 12.19 | 23.49 | 11.43 |
| NNGen | 15.97 | 28.14 | 13.82 | 5.74 | 16.33 | 7.18 | 12.76 | 23.93 | 11.58 |
| CODE-NN | 10.08 | 26.17 | 11.33 | 3.86 | 15.25 | 6.19 | 8.24 | 22.28 | 9.61 |
| Hybrid-DRL | 9.29 | 30.00 | 12.47 | 6.30 | 24.19 | 10.30 | 8.42 | 28.64 | 11.73 |
| Transformer | 12.91 | 28.04 | 13.83 | 5.75 | 18.62 | 9.89 | 10.69 | 24.65 | 12.02 |
| Dual Model | 11.49 | 29.20 | 13.24 | 5.25 | 21.31 | 9.14 | 9.61 | 26.40 | 11.87 |
| Rencos | 14.80 | 31.41 | 14.64 | 7.54 | 23.12 | 10.35 | 12.59 | 28.45 | 13.21 |
| GCN2Seq | 9.79 | 26.59 | 11.65 | 4.06 | 18.96 | 7.76 | 7.91 | 23.67 | 10.23 |
| GAT2Seq | 10.52 | 26.17 | 11.88 | 3.80 | 16.94 | 6.73 | 8.29 | 22.63 | 10.00 |
| SeqGNN | 10.51 | 29.84 | 13.14 | 4.94 | 20.80 | 9.50 | 8.87 | 26.34 | 11.93 |
| *HGNN w/o augment & static* | 11.75 | 29.59 | 13.86 | 5.57 | 22.14 | 9.41 | 9.98 | 26.94 | 12.05 |
| *HGNN w/o augment & dynamic* | 11.85 | 29.51 | 13.54 | 5.45 | 21.89 | 9.59 | 9.93 | 26.80 | 12.21 |
| *HGNN w/o augment* | 12.33 | 29.99 | 13.78 | 5.45 | 22.07 | 9.46 | 10.26 | 27.17 | 12.32 |
| *HGNN w/o static* | 15.93 | 33.67 | 15.67 | 7.72 | 24.69 | 10.63 | 13.44 | 30.47 | 13.98 |
| *HGNN w/o dynamic* | 15.77 | 33.84 | 15.67 | 7.64 | 24.72 | 10.73 | 13.31 | 30.59 | 14.01 |
| ***HGNN*** | **16.72** | **34.29** | **16.25** | **7.85** | **24.74** | **11.05** | **14.01** | **30.89** | **14.50** |

Similar to previous works (Zhang et al., 2020; Wan et al., 2018; Fernandes et al., 2018; Iyer et al., 2016), BLEU (Papineni et al., 2002), METEOR (Banerjee & Lavie, 2005) and ROUGE-L (Lin, 2004) are used as our automatic evaluation metrics. These metrics are popular for evaluating machine translation and text summarization tasks. Except for these automatic metrics, we also conduct a human evaluation study. We invite 5 PhD students and 10 master students as volunteers, who have rich C programming experiences. The volunteers are asked to rank summaries generated from the anonymized approaches from 1 to 5 (*i.e.*, 1: Poor, 2: Marginal, 3: Acceptable, 4: Good, 5: Excellent) based on the relevance of the generated summary to the source code and the degree of similarity between the generated summary and the actual summary. Specifically, we randomly choose 50 functions for each model with the corresponding generated summaries and ground-truths. We calculate the average score and the higher the score, the better the quality.

## 3.2 COMPARISON WITH THE BASELINES

Table 1 shows the evaluation results including two parts: the comparison with baselines and the ablation study. Consider the comparison with state-of-the-art baselines, in general, we find that our proposed model outperforms existing methods by a significant margin on both in-domain and out-of-domain datasets, and shows good generalization performance. Compared with others, on in-domain dataset, the retrieval-based approaches could achieve competitive performance on BLEU-4, however ROUGE-L and METEOR are fare less than ours. Moreover, they do not perform well on the out-of-domain dataset. Compared with the graph-based approaches (*i.e.*, GCN2Seq, GAT2Seq and SeqGNN), even without augmentation (*HGNN w/o augment*), our approach still outperforms them, which further demonstrates the effectiveness of Hybrid GNN for additionally capturing global graph information. Compared with Rencos that also considers the retrieved information in the Seq2Seq model, its performance is still lower than *HGNN*. On the overall dataset including both of in-domain and out-of-domain data, our model achieves **14.01**, **30.89** and **14.50**, outperforming current state-of-the-art method Rencos by **1.42**, **2.44** and **1.29** in terms of BLEU-4, ROUGE-L and METEOR metrics.

## 3.3 ABLATION STUDY

We also conduct an ablation study to evaluate the impact of different components of our framework, *e.g.*, retrieval-based augmentation, static graph and dynamic graph in the last row of Table 1. Overall, we found that 1) retrieval-augmented mechanism could contribute to the overall model performance (*HGNN* vs. *HGNN w/o augment*). Compared with *HGNN*, we see that the performance of *HGNN w/o static* and *HGNN w/o dynamic* decreases, which demonstrates the effectiveness of the Hybrid GNN and 2) the performance without static graph is worse than the performance without dynamic graph in ROUGE-L and METEOR, however, BLEU-4 is higher than the performance without dynamic graph. To further understand the impact of the static graph and dynamic graph, we evaluate the performance without augmentation and static graph/dynamic graph (see *HGNN w/o augment& static* and *HGNN w/o augment& dynamic*). Compared with *HGNN w/o augment*, the results further confirm the effectiveness of the Hybrid GNN (*i.e.*, static graph and dynamic graph).

Table 2: Human evaluation results on the CCSD test set.

| Metrics | NNGen | Transformer | Rencos | SeqGNN | *HGNN* |
|---|---|---|---|---|---|
| Relevance | 3.23 | 3.17 | 3.48 | 3.09 | **3.69** |
| Similarity | 3.18 | 3.02 | 3.32 | 3.06 | **3.51** |

Table 3: Examples of generated summaries on the CCSD test set.

| Example | Example 1 | Example 2 |
|---|---|---|
| Source Code | ```static void strInit(Str *p){ p->z = 0; p->nAlloc = 0; p->nUsed = 0; }``` | ```void ReleaseCedar(CEDAR *c){ if (c == NULL) return; if (Release(c->ref) == 0) CleanupCedar(c); }``` |
| Ground-Truth | initialize a str object | release reference of the cedar |
| NNGen | free the string | release the virtual host |
| Transformer | reset a string | release of the cancel object |
| Rencos | append a raw string to the json string | release of the cancel object |
| SeqGNN | initialize the string | release cedar communication mode |
| *HGNN* | **initialize a string object** | **release reference of cedar** |

We also conduct experiments to investigate the impact of code-based augmentation and summary-based augmentation. Overall, we found that the summary-based augmentation could contribute more than the code-based augmentation. For example, after adding the code-based augmentation, the performance can be 10.22, 27.54 and 12.49 in terms of BLUE-4, ROUGE-L and METEOR on the overall dataset. With the summary-based augmentation, the results can reach to 13.76, 30.59 and 14.11. Compared with the results without augmentation (*i.e.*, 10.26. 27.17, 12.32 with *HGNN w/o augment*), we can see that code-based augmentation could have some improvement, but the effect is not significant compared with summary-based augmentation. We conjecture that, due to that the code and summary are heterogeneous, the summary-based augmentation has a more direct impact on the code summarization task. When combining both code-based augmentation and summary-based augmentation, we can achieve the best results (*i.e.*, 14.01, 30.89, 14.50). We plan to explore more code-based augmentation (*e.g.*, semantic-equivalent code transformation) in our future work.

## 3.4 HUMAN EVALUATION

As shown in Table 2, we perform a human evaluation on the overall dataset to assess the quality of the generated summaries by our approach, NNGen, Transformer, Rencos and SeqGNN in terms of relevance and similarity. As depicted in Table 1, NNGen, Rencos and SeqGNN are the best retrieval-based, sequence-based, and graph-based approaches, respectively. We also compare with Transformer as it has been widely used in natural language processing. The results show that our method can generate better summaries which are more relevant with the source code and more similar with the ground-truth summaries.

## 3.5 CASE STUDY

To perform qualitative analysis, we present two examples with generated summaries by different methods from the overall data set, shown in Table 3. We can see that, in the first example, our approach can learn more code semantics, *i.e.*, $p$ is a self-defined struct variable. Thus, we could generate a token *object* for the variable $p$. However, other models can only produce *string*. Example 2 is a more difficult function with the functionality to "release reference of cedar", as compared to other baselines, our approach effectively captures the functionality and generates a more precise summary.

## 3.6 EXTENSION ON THE PYTHON DATASET

We conducted additional experiments on a public dataset, *i.e.*, the Python Code Summarization Dataset (PCSD), which was also used in Rencos (the most competitive baseline in our paper). We follow the setting of Rencos and split PCSD into the training set, validation set and testing set with fractions of 60%, 20% and 20%. We construct the static graph based on AST. The decoding step is set to 50, followed by Rencos, and the other settings are the same with CCSD. We compare our methods on PCSD against various competitive baselines, *i.e.*, NNGen, CODE-NN, Rencos and

Table 4: Automatic evaluation results (in %) on the PCSD test set.

| Methods | BLEU-4 | ROUGE-L | METEOR |
|---|---|---|---|
| NNGen | 21.60 | 31.61 | 15.96 |
| CODE-NN | 16.39 | 28.99 | 13.68 |
| Transformer | 17.06 | 31.16 | 14.37 |
| Rencos | 24.02 | 36.21 | 18.07 |
| *HGNN w/o static* | 24.06 | 38.28 | 18.66 |
| *HGNN w/o dynamic* | 24.13 | 38.64 | 18.93 |
| ***HGNN*** | **24.42** | **39.91** | **19.48** |

Transformer, which are either retrieval-based, generation-based or hybrid methods. The results are shown in Table 4. The results indicate that, compared with the best results from NNGen, CODE-NN, Rencos and Transformer, our method can improve the performance by 0.40, 3.70 and 1.41 in terms of BLEU-4, ROUGE-L and METEOR. We also conduct the ablation study on PCSD to demonstrate the usefulness of the static graph (*i.e.*, HGNN w/o dynamic) and dynamic graph (*i.e.*, HGNN w/o static). The results also demonstrate that both static graph and dynamic graph can contribute to our framework. In summary, the results on both our released benchmark (CCSD) and existing benchmark (PCSD) demonstrate the effectiveness and the scalability of our method.

## 4    RELATED WORK

**Source Code Summarization** Early works (Eddy et al., 2013; Haiduc et al., 2010; Wong et al., 2015; 2013) for code summarization focused on using information retrieval to retrieve summaries. Later works attempted to employ attentional Seq2Seq model on the source code (Iyer et al., 2016; Siow et al., 2020) or some variants, i.e., AST (Hu et al., 2018a; Alon et al., 2018; Liu et al., 2020) for generation. However, these works are based on sequential models, ignoring rich code semantics. Some latest attempts (LeClair et al., 2020; Fernandes et al., 2018) embedded program semantics into GNNs. but they mainly rely on simple representations, which are limited to learn full semantics.

**Graph Neural Networks** Over the past few years, GNNs (Li et al., 2015; Hamilton et al., 2017; Kipf & Welling, 2016; Chen et al., 2020b) have attracted increasing attention with many successful applications in computer vision (Norcliffe-Brown et al., 2018), natural language processing (Xu et al., 2018a; Chen et al., 2020d;c;e). Because by design GNNs can model graph-structured data, recently, some works have extended the widely used Seq2Seq architectures to Graph2Seq architectures for various tasks including machine translation (Beck et al., 2018), and graph (e.g., AMR, SQL)-to-text generation (Zhu et al., 2019; Xu et al., 2018b). Some works have also attempted to encode programs with graphs for diverse tasks e.g., VARNAMING/VARMISUSE (Allamanis et al., 2017), Source Code Vulnerability Detection (Zhou et al., 2019). As compared to these works, we innovate a hybrid message passing GNN performed on both static graph and dynamic graph for message fusion.

## 5    CONCLUSION AND FUTURE WORK

In this paper, we proposed a general-purpose framework for automatic code summarization. A novel retrieval-augmented mechanism is proposed for combining the benefits of both retrieval-based and generation-based approaches. Moreover, to capture global semantics among nodes, we develop a hybrid message passing GNN based on both static and dynamic graphs. The evaluation shows that our approach improves state-of-the-art techniques substantially. Our future work includes: 1) we plan to introduce more information such as API knowledge to learn the better semantics of programs, 2) we explore more code-based augmentation techniques to improve the performance and 3) we plan to adopt the existing techniques such as (Du et al., 2019; Xie et al., 2019a;b; Ma et al., 2018) to evaluate the robustness of the trained model.

## 6    ACKNOWLEDGMENTS

This research is partially supported by the National Research Foundation, Singapore under its the AI Singapore Programme (AISG2-RP-2020-019), the National Research Foundation, Prime Ministers Office, Singapore under its National Cybersecurity R&D Program (Award No. NRF2018NCR-NCR005-0001), NRF Investigatorship NRF-NRFI06-2020-0001, the National Research Foundation through its National Satellite of Excellence in Trustworthy Software Systems (NSOE-TSS) project under the National Cybersecurity R&D (NCR) Grant award no. NRF2018NCR-NSOE003-0001.

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

# Appendices

## A DETAILS ON CODE PROPERTY GRAPH

Code Property Graph (CPG) (Yamaguchi et al., 2014), which is constructed on abstract syntax tree (AST), combines different edges (i.e., Flow to, Control) to represent the semantics of the program. We describe each representation combining with Figure 2 as follows:

- **Abstract Syntax Tree (AST).** AST contains syntactic information for a program and omits irrelevant details that have no effect on the semantics. Figure 2 shows the completed AST nodes on the left simple program and each node has a code sequence in the first line and type attribute in the second line. The black arrows represent the child-parent relations among ASTs.

- **Control Flow Graph (CFG).** Compared with AST highlighting the syntactic structure, CFG displays statement execution order, i.e., the possible order in which statements may be executed and the conditions that must be met for this to happen. Each statement in the program is treated as an independent node as well as a designated entry and exit node. Based on the keywords *if*, *for*, *goto*, *break* and *continue*, control flow graphs can be easily built and "Flow to" with green dashed arrows in Figure 2 represents this flow order.

- **Program Dependency Graph (PDG).** PDG includes **data dependencies** and **control dependencies**: 1) data dependencies are described as the definition of a variable in a statement reaches the usage of the same variable at another statement. In Figure 2, the variable "*b*" is defined in the statement "*int b = a++*" and used in "*call (b)*". Hence, there is a "Reach" edge with blue arrows point from "*int b = a++*" to "*call (b)*". Furthermore, Define/Use edge with orange double arrows denotes the definition and usage of the variable. 2) different from CFG displaying the execution process of the complete program, control dependencies define the execution of a statement may be dependent on the value of a predicate, which more focus on the statement itself. For instance, the statements "*int b = a++*" and "*call(b)*" are only performed "if a is even". Therefore, a red double arrow "Control" points from "*if (a % 2) == 0*" to "*int b = a++*" and "*call(b)*".

## B DETAILS ON BASELINE METHODS

We compare our approach with existing baselines. They can be divided into three groups: Retrieval-based approaches, Sequence-based approaches and Graph-based approaches. For papers that provide the source code, we directly reproduce their methods on CCSD dataset. Otherwise, we reimplement their approaches with reference to the papers.

### B.1 RETRIEVAL-BASED APPROACHES

**TF-IDF** (Haiduc et al., 2010) is the abbreviation of Term Frequency-Inverse Document Frequency, which is adopted in the early code summarization (Haiduc et al., 2010). It transforms programs into weight vectors by calculating term frequency and inverse document frequency. We retrieve the summary of the most similar programs by calculating the cosine similarity on the weighted vectors.

**NNGen** (Liu et al., 2018) is a retrieved-based approach to produce commit messages for code changes. We reproduce such an algorithm on code summarization. Specifically, we retrieve the most similar top-k code snippets on a bag-of-words model and prioritizes the summary in terms of BLEU-4 scores in top-k code snippets.

### B.2 SEQUENCE-BASED APPROACHES

**CODE-NN** (Iyer et al., 2016; Barone & Sennrich, 2017) adopts an attention-based Seq2Seq model to generate summaries on the source code.

**Transformer** (Ahmad et al., 2020) adopts the transformer architecture (Vaswani et al., 2017) with self-attention to capture long dependency in the code for source code summrization.

**Hybrid-DRL** (Wan et al., 2018) is a reinforcement learning-based approach, which incorporates AST and sequential code snippets into a deep reinforcement learning framework and employ evaluation metrics e.g., BLEU as the reward.

**Dual Model** (Wei et al., 2019) propose a dual training framework by training code summarization and code generation tasks simultaneously to boost each task performance.

**Rencos** (Zhang et al., 2020) is the retrieval-based Seq2Seq model for code summarization. it utilized a pretrained Seq2Seq model during the testing phase by computing a joint probability conditioned on both the original source code and retrieved the most similar source code for the summary generation. Compared with Rencos, we propose a novel retrieval-augmented mechanism for the similar source code and use it at the model training phase.

### B.3 GRAPH-BASED APPROACHES

We also compared with some latest GNN-based works, employing graph neural network for source code summarization.

**GCN2Seq, GAT2Seq** modify Graph Convolution Network (Kipf & Welling, 2016) and Graph Attention Network (Velickovic et al., 2018) to perform convolution operation and attention operation on the code property graph for learning and followed by a LSTM to generate summaries. We implement the related code from scratch.

**SeqGNN** (Fernandes et al., 2018) combines GGNNs and standard sequence encoders for summarization. They take the code and relationships between elements of the code as input. Specially, a BiLSTM is employed on the code sequence to learn representations and each source code token is modelled as a node in the graph, and employed GGNN for graph-level learning. Since our node sequences are sub-sequence of source code rather than individual token, we adjust to slice the output of BiLSTM and sum each token representation in node sequences as node initial representation for summarization. Furthermore, we implement the related code from scratch.

## C MODEL SETTINGS

We embed the most frequent 40,000 words in the training set with 512-dims and set the hidden size of BiLSTM to 256 and the concatenated state size for both directions is 512. The dropout is set to 0.3 after the word embedding layer and BiLSTM. We set GNN hops to 1 for the best performance. The optimizer is selected with Adam with an initial learning rate of 0.001. The batch size is set to 64 and early stop for 10. The beam search width is set to 5 as usual. All experiments are conducted on the DGX server with four Nvidia Graphics Tesla V100 and each epoch takes 6 minutes averagely. All hyperparameters are tuned with grid search on the validation set.

## D DETAILS ON DATA PREPARATION

It is non-trivial to obtain high-quality datasets for code summarization. We noticed that despite some previous works (Barone & Sennrich, 2017; Hu et al., 2018b) released their datasets, however, they are all based on high-level programming languages i.e. Java, Python. We are the first to explore summarization on *C* programming language. Specifically, we crawled from popular *C* repositories (e.g., Linux and QEMU) on GitHub, and then extracted separate function-summary pairs from these projects. Specifically, we extracted functions and associated comments marked by special characters "/**" and "*/" over the function declaration. These comments can be considered as explanations of the functions. We filtered out functions with line exceeding 1000 and any other comments inside the function, and the first sentence was selected as the summary. A similar practice can be found in (Jiang et al., 2017). Totally, we collected **500k+** raw function-summary pairs. Furthermore, functions with token size greater than 150 were removed for computational efficiency and there were **130k+** functions left. Since duplication is very common in existing datasets (Fernandes et al., 2018), followed by Allamanis (2019), we performed a de-duplication process and removed functions with similarity over 80%. Specifically, we calculated the cosine similarity by encoding the raw functions into vectors with sklearn. Finally, we kept **95k+** unique functions. We name this dataset *C* Code Summarization Dataset (CCSD). To testify model generalization ability, we randomly selected

Table 5: More Examples of generated summaries on the CCSD test set.

| Example | Example 1 | Example 2 |
|---|---|---|
| Source Code | ```static void counterMutexFree (sqlite3_mutex *p){ assert(g.isInit); g.m.xMutexFree(p->pReal); if( p->eType==SQLITE_MUTEX_FAST || p->eType== S   QLITE_MUTEX_RECURSIVE) { free(p); } }``` | ```static void __exit wimax_subsys_exit(void) { wimax_id_table_release(); genl_unregister_family (&wimax_gnl_family); }``` |
| Ground-Truth | free a countable mutex | shutdown the wimax stack |
| NNGen | enter a countable mutex | unregisters pmcraid event family return value none |
| Transformer | leave a mutex | de initialize wimax driver |
| Rencos | try to enter a mutex | unregister the wimax device subsystem |
| SeqGNN | free a mutex allocated by sqlite3 mutex | this function is called when the driver is not held |
| *HGNN* | **release a mutex** | **free the wimax stack** |
| Retrieved_code | ```static int counterMutexTry (sqlite3_mutex *p){ assert( g.isInit ); assert( p->eType>=0 ); assert( p->eType<MAX_MUTEXES ); g.aCounter[p->eType]++; if( g.disableTry ) return SQLITE_BUSY; return g.m.xMutexTry(p->pReal); }``` | ```static int __init wimax_subsys_init(void){ int result; d_fnstart(4, NULL, "()\n"); d_parse_params(D_LEVEL, D_LEVEL_SIZE, wimax_debug_params, "wimax.debug"); result = genl_register_family (&wimax_gnl_family); if (unlikely(result < 0)) { pr_err("cannot register generic netlink family: %d\n", result); goto error_register_family;} d_fnend(4, NULL, "() = 0\n"); return 0; error_register_family: d_fnend(4, NULL, "() = %d\n", result); return result; }``` |
| Retrieved_summary | try to enter a mutex | shutdown the wimax stack |
| Example | Example 3 | Example 4 |
| Source Code | ```static void udc_dd_free( struct lpc32xx_udc *udc, struct lpc32xx_usbd_dd_gad *dd) { dma_pool_free(udc->dd_cache, dd, dd->this_dma); }``` | ```void ReleaseSockEvent(SOCK_EVENT *event) { if (event == NULL) { return; } if (Release(event->ref) == 0) { CleanupSockEvent(event); } }``` |
| Ground-Truth | free a dma descriptor | release of the socket event |
| NNGen | allocate a dma descriptor | clean up of the socket event |
| Transformer | free the usb device | set the event |
| Rencos | allocate a dma descriptor | set of the sock event |
| SeqGNN | free dma buffers | release of the socket |
| *HGNN* | **free a dma descriptor** | **release the sock event** |
| Retrieved_code | ```static struct lpc32xx_usbd_dd_gad *udc_dd_alloc(struct lpc32xx_udc *udc) { dma_addr_t   dma; struct lpc32xx_usbd_dd_gad *dd; dd = dma_pool_alloc(udc->dd_cache, GFP_ATOMIC | GFP_DMA, &dma); if (dd) dd->this_dma = dma; return dd; }``` | ```void SetL2TPServerSockEvent( L2TP_SERVER *l2tp,SOCK_EVENT *e){ if (l2tp == NULL) { return;} if (e != NULL){ AddRef(e->ref);} if (l2tp->SockEvent != NULL){ ReleaseSockEvent(l2tp->SockEvent); l2tp->SockEvent = NULL;} l2tp->SockEvent = e;}``` |
| Retrieved_summary | allocate a dma descriptor | set a sock event to the l2tp server |

some projects as the out-of-domain test set with 2,330 examples and the remaining were randomly split into train/validation/test with 84,316/4,432/4,203 examples. The open-source code analysis platform Joern (Yamaguchi et al., 2014) was applied to construct the code property graph.

# E   MORE EXAMPLES

We show more examples along with the retrieved code and summary by dynamic programming in Table 5 and we can find that *HGNN* can generate more high-quality summries based on our approach.

