# OpenReview forum: "Retrieval-Augmented Generation for Code Summarization via Hybrid GNN"
_ICLR.cc/2021/Conference — ICLR 2021 Spotlight_

### Official Review · AnonReviewer3 · 2020-10-28
**Good Results and More Evaluation Needed**

**Rating:** 7
**Confidence:** 3

**Review:**


Overview:

The authors tackle the code summarization problem. They retrieve the most similar code and use it as additional input features. They also using GNN to get features from static and dynamic graphs. They evaluate their results on their collected C projects (C-Code-Summarization Benchmark) with 1-2% improvement on automatic evaluation.


Reasons to accept:
* They collect and release a new challenging C benchmark for code summarization.
* They propose a hybrid-GNN solution to capture global graph information.

Reasons to reject:
* No evaluation on any publicly available datasets. Even though there are some issues about duplication, I will still expect to see such a comparison with other baselines.

* Attention-based dynamic message passing of graph model is not proposed in this paper, and I don't think it is necessary to have this "hybrid" design, maybe only dynamic one is enough. As shown in Table 1, the dynamic is more important than static (Although we do see an overall performance, except the out-of-domain meteor score, using both are still better, but it is marginal).


Questions & Suggestions:
* Can you provide more dataset information? For example, the average lines of the code, the average length of the natural language summarization. It seems to me from the examples that each example only has few lines of code and the summarization is very short, it is more like a topic modeling task.

* To my understanding, this work is not the first work combining retrieval solution with generation model for code summarization (e.g., Retrieval-based Neural Source Code Summarization) so please modify some of the claims in the paper.

* Can you provide more details about the human evaluation? What is the agreement value?

* "z is the similarity score, which is introduced to weaken the negative impact of c′ on the original training data c", how do you find the best c?

* When you run baselines on your dataset, did you do a hyper-parameter search or just use their default setting (especially the Rencos model)?

* There are some retrieved-augment language models in the NLP field that the authors may want to take a look and compare with, for example, "Retrieval-Augmented Generation for Knowledge-Intensive NLP Tasks".

* Did you run any baselines that are based on pre-trained language model, such as BERT, BART, T5, or even more code related like CodeBERT: A Pre-Trained Model for Programming and Natural Languages?

---

> ### Author Response · Authors · 2020-11-21
> **Author response to Review #3-Part 1**
>
> Please refer to A1 in the common responses for the evaluation on a public dataset. Besides, we thank the reviewer on other detailed comments and we addressed these comments as follows:
>
> Q1: Attention-based dynamic message passing of graph model is not proposed in this paper, and I don't think it is necessary to have this "hybrid" design, maybe only dynamic one is enough. As shown in Table 1, the dynamic is more important than static (Although we do see an overall performance, except the out-of-domain meteor score, using both are still better, but it is marginal).
>
> A1: We agree that the idea of using the dynamic graph with GNNs is not our contribution since prior works have explored this idea as well. Our contributions mainly include: 1) the retrieval-based augmentation for generation models and 2) the Hybrid GNN leveraging both static and dynamic graphs. We will make the statement on contributions more clear in the revision. We also added more ablation study results for the retrieval-based augmentation. Please refer to A2 in the common response.
>
> As for the Hybrid GNN, although overall the dynamic graph performs better than the static graph in our experiments, we think the static graph is still needed and useful: 1)  Considering the complexity of this task and size of the testing set (6388 samples in CCSD and 21028 in PCSD), we think the performance improvement of HGNN (i.e., using both static and dynamic graphs) compared with HGNN w/o static and HGNN w/o dynamic is still promising. 2) There are still some results showing that the static graph achieves better performance than the dynamic graph. Please see the results a) BLUE-4 values (i.e., 12.00 vs 11.87) between (HGNN w/o augment & dynamic) and (HGNN w/o augment & static) and b) The METEOR results in the new added python results in Q1 of common response, i.e., the METEOR values of HGNN w/o dynamic and HGNN w/o static are 18.42 and 18.36, respectively. 3) Moreover, we think GNN with hybrid static and dynamic graphs is a promising idea in general, and we believe researchers working on other domain applications might find it interesting and helpful.
>
> Q2: Can you provide more dataset information? For example, the average lines of the code, the average length of the natural language summarization. It seems to me from the examples that each example only has few lines of code and the summarization is very short, it is more like a topic modeling task.
>
> A2: For our C dataset, the average lines of the code are 12.59 and the average token length of summary is 8.22. For the new added dataset PCSD, the average lines of the code are 14.24 and the average token length is 10.91. Although the code summarization task and the topic modeling task share some kind of similarity, we think they are still quite different. First of all, the summarization task usually aims to provide a more fine-grained description of the input data while topic modeling aims to provide some high-level description (e.g., keywords) of the themes of input data. Secondly, different from topic modeling which usually takes as input free-form text and outputs a few keywords to describe its themes, the key challenge of code summarization compared to regular text summarization is that the input (i.e., source code) and the expected output (i.e., summary) are from two very different domains, i.e., they are heterogeneous data. To solve this challenge, the state-of-the-art techniques adopt the deep neural networks to learn the code semantics and generate the summary. Our work follows this line of research and proposes a novel method, i.e., the retrieval-augmented hybrid GNN.
>
> Q3: To my understanding, this work is not the first work combining retrieval solution with generation model for code summarization (e.g., Retrieval-based Neural Source Code Summarization) so please modify some of the claims in the paper.
>
> A3: Thanks for the suggestions! We agree that this is not the first work that proposes the retrieval-generation method for code summarization. And we did cite the Retrieval-based Neural Source Code Summarization work (which is the Rencos baseline in our experiments) in our paper. In fact, different from Rencos, which feeds the combination of the retrieved code and the test code to a seq2seq model, we propose a novel retrieval augment mechanism to employ the similar code and its summary for model training and encode more program semantics with GNN for the summary generation. We will make our claim more clear and add more discussions on the differences in the revision.

---

> > ### Author Response · Authors · 2020-11-21
> > **Author response to Review #3-Part 2**
> >
> > Due to the word limit for each comment, we add the response to the remaining comments.
> >
> > Q4: Can you provide more details about the human evaluation? What is the agreement value?
> >
> > A4: We asked 15 volunteers to evaluate the similarity and the relevance (scoring form 1-5) between the generated summary and the ground truth for each test example. We further calculated the average ratings of the 15 volunteers. The standard deviation of the similarity and relevance scores are 0.38 and 0.30, which demonstrates that the volunteers have a high agreement.
> >
> > Q5: z is the similarity score, which is introduced to weaken the negative impact of c′ on the original training data c, how do you find the best c?
> >
> > A5: The reviewer might ask how to select the best c’, i.e., the retrieved source code, based on the original source code c. Actually, z is the text similarity score (i.e., z=sim(c,c’)), please see sim(c,c’) in step 1. Note that for each training data c, we will select the best c’, where c’ is the candidate (in D’) that has the highest similarity score with c (i.e., the largest z). We will add a more clear description in the revision.
> >
> > Q6: When you run baselines on your dataset, did you do a hyper-parameter search or just use their default setting (especially the Rencos model)?
> >
> > A6: We did tune hyperparameters for baseline methods in our experiments. Specifically, we customize the max length of the input and output based on our CCSD. Furthermore, we did a hyper-parameter search including embedding size, learning rate of the baselines to select the best configuration for each baseline for a fair comparison.
> >
> > Q7: There are some retrieved-augment language models in the NLP field that the authors may want to take a look and compare with, for example, "Retrieval-Augmented Generation for Knowledge-Intensive NLP Tasks".
> >
> > A7:  Thanks for pointing out the related paper. Actually, there are two main differences: 1) the applications are different. We focus on code summarization that requires the better semantic learning of programs (e.g., how to better learn semantics from the existing structures such as AST, PDG ), throwing different challenges with NLP tasks and 2) Since the gap between the code and summary, focus on the problem, we propose a novel retrieval augment mechanism to combine the retrieved information into GNN-based generation model for summary generation, which is different from the mentioned paper, combining a pre-trained retriever (Query Encoder and Document Index) with a pre-trained encoder-decoder (generator) for knowledge-intensive tasks.
> >
> > Q8: Did you run any baselines that are based on pre-trained language model, such as BERT, BART, T5, or even more code related like CodeBERT: A Pre-Trained Model for Programming and Natural Languages?
> >
> > A8: We added new experiments of CodeBert. Based on the released pre-trained model [1], we finetune the model for the code summarization task on CCSD. The experimental results on CCSD (Overall test set) are shown below. Compared with the large-scale pre-trained model CodeBert, HGNN achieves a comparable performance. Notably, we argue that this is not a fair comparison since CodeBert employs 6 programming languages i.e., Java, JavaScript, PHP, Python, Go and Ruby, with a total of 2,137,293 samples for pretraining. In addition, much more GPU resources (16 V100 GPUs) are required to train CodeBert (10 hours per epoch). Differently, we did not do any kind of pretraining on large datasets, and our model can be quickly trained on one GPU (6 minutes per epoch). In summary, the problem settings and resource requirements of the two works are quite different. However, our approach can still achieve a competitive performance, demonstrating its effectiveness in the code summarization task.
> >
> > Methods \ Metrics | BLEU-4 | ROUGE-L | METEOR
> >
> > CodeBert | 10.56 | 29.38 | 32.99
> >
> > HGNN | 13.39 | 30.23 | 26.22
> >
> > [1]. CodeXGLUE  https://github.com/microsoft/CodeXGLUE

---

### Official Review · AnonReviewer4 · 2020-10-28
**This paper leverages similar codes to help generate code summarization, and an attention-based dynamic graph model is introduced to further capture the global graph information.**

**Rating:** 7
**Confidence:** 3

**Review:**

Summary:

This paper leverages similar code-summary pairs from existing data to assist code summary generation. The model first retrieves a similar code snippet from the existing database. Then, the author applied GNN over the code property graphs (CPGs).  A challenge is that CPGs are typically deep therefore it is difficult to capture long dependencies. The author proposed an attention mechanism to capture global information between nodes, and then a hybrid GNN layer encodes the retrieve-augmented graph.  Finally, a generator takes both GNN's output and the retrieved text summary and predict outputs. Experimental results over a new C code indicates that the proposed method outperforms both IR and neural generation methods.

########################################

Reason for score:

Overall, I vote for accepting. Both the idea of leveraging existing code and also the adaptive layer to capture long dependencies are interesting and the experiments look solid. Although I would still like to see the results from previous existing datasets.

########################################
Some comments about the experiments:

a. As an application study, it is still necessary to compare the model over previous benchmarks, even though there are some issues with those datasets.

b. A pair of missing ablation studies are: a generator still takes the text summary of retrieved code, but not use the augmented graph; and vice versa, the generator only takes the graph information but not the retrieved text summary. This can further indicate which part of the retrieved information is more useful.

---

> ### Author Response · Authors · 2020-11-21
> **Author response to Review #4**
>
> We thank the reviewer again for the useful comments.
> Please refer to A1 and A2 in the common responses for the evaluation results on a public dataset and the ablation study with only code-based augmentation and only summary-based augmentation.

---

### Official Review · AnonReviewer1 · 2020-10-29
**Retrieval-augmented code summarization model with state-of-the-art results on a newly curated dataset**

**Rating:** 7
**Confidence:** 3

**Review:**

Summary

This paper proposes a retrieval-augmented method for generating code summarization. The model encodes the input code based on its graph structure (Code Property Graph) with a hybrid GNN architecture. The model augments the initial graph representation of the input code based on the representation of the top-1 retrieval result. It also augments the final graph encoding with the BiLSTM encoding of the retrieved summary.

The proposed model is evaluated on a newly curated C code summarization data set and shows state of the art performance compared against previous systems.

Strengths
- Releases a new C code summarization data, which will be beneficial for the community.
- This paper reports human evaluation results.
- Ablation study shows the retrieval augmentation and the new hybrid GNN architecture is helpful.

Weaknesses
- The model is not evaluated on any existing code summarization benchmarks. Showing that the proposed architecture is generally applicable by getting good results on more benchmarks will make the story a lot more convincing.
- Could include more analysis (see below for details)

Other questions/comments
- Would be nice to provide the mathematical formulation of “Step 3: Retrieved Summary-based Augmentation”
- Would it be possible to ablate code-based summarization vs. summary-based augmentation separately? It would be interesting to see their relative impact.
- Suggestion: it would be helpful to provide an input-output example earlier in the paper.
- Suggestion: would be nice to include real examples of retrieval results in the analysis section.
- Have you considered using top-k retrieval results instead of top-1?

---

> ### Author Response · Authors · 2020-11-21
> **Author response to Review #1**
>
> Please see the evaluation results on other dataset and the ablation study on Retrieval-based Augmentation in the common response.
>
> Other questions and comments:
> We thank the reviewer for the detailed comments again and we address them in the revision as follows:
>
> Q1: Would be nice to provide the mathematical formulation of “Step 3: Retrieved Summary-based Augmentation”
>
> A1: We provide the formula as follows in the revision:
>
> We further encode the retrieved summary $s'$ with another BiLSTM model. We represent each token $t'_i$ of $s'$ using the learned embedding matrix $\boldsymbol E^{seqtoken}$. Then $s'$ can be encoded as:
>
>
> \begin{equation}
> \boldsymbol h_{t_1'},...,\boldsymbol h_{t_T'} = \mathrm{BiLSTM}(E^{seqtoken}_{t_1'}  ,...,  E^{seqtoken}_{t_T'})
> \end{equation}
>
>
> where $ h_{t'_i}$ is the state of the BiLSTM model for the token $t_i'$ in $s'$ and $T$ is the length of $s'$. We also multiply the similarity score $z$ to $[\boldsymbol h_{t_1'},...,\boldsymbol h_{t_T'}]$  and concatenate with the graph encoding results (i.e., the outputs of the GNN encoder) as the input $[\mathrm{GNN}, z \boldsymbol h_{t_1'},...,z \boldsymbol h_{t_T'}]$  to the decoder.
>
>
>
> Q2: Suggestion: it would be helpful to provide an input-output example earlier in the paper.
>
> A2: We will add an illustrative example (including the input and the expected summary results) in our paper such that the readers could better understand our method.
>
> Q3: Suggestion: would be nice to include real examples of retrieval results in the analysis section.
>
> A3: We will follow the suggestion and add concrete cases for illustrating the retrieval results in our revision. For example, one concrete example (Example 2 in Table 3) is shown as follows:
>
> Input code:
>
> void ReleaseCedar(CEDAR *c) {
>
>        if (c == NULL)
>         {
>                 return;
>         }
>
>         if (Release(c->ref) == 0)
>         {
>                 CleanupCedar(c);
>         }
> }
>
> Ground-Truth: release reference of the cedar.
>
> Retrieved code:
>
> void DelConnection(CEDAR *cedar, CONNECTION *c)
>
> {
>
> 	if (cedar == NULL || c == NULL)
>
> 	{
>
> 		return;
>
> 	}
>
> 	LockList(cedar->ConnectionList);
>
> 	{
>
> 		Debug("Connection %s Deleted from Cedar.\n", c->Name);
>
> 		if (Delete(cedar->ConnectionList, c))
>
> 		{
>
> 			ReleaseConnection(c);
>
> 		}
>
> 	}
>
> 	UnlockList(cedar->ConnectionList);
>
> }
>
> Retrieved summary: delete connection from cedar.
>
> Our result (HGNN): release reference of cedar.
>
> We conjecture that the reason our method can generate a high-quality summary for this example could probably be that: With the retrieved summary and code, our method might be able to learn the mapping pattern between the method name and summary. For example, DelConnection (CEDAR* cedar, CONNECTION *c)  -> “delete connection from cedar“. Then for this test example ReleaseCedar(CEDAR *c), our method might learn to leverage this pattern and get the correct result.
>
> Q4: Have you considered using top-k retrieval results instead of top-1?
>
> A4: Thanks for this great suggestion! In fact, we also explored the top-2 and top-3 retrieval results with our method. However, there was no significant performance boost when more retrieval results were used. Moreover, more GPU resources were needed for expensive training. Thus, we only consider the top-1 retrieval result in this work. We will add more discussion on the effect of different top-k retrieval results in the revision. And we will leave how to effectively utilize top-k retrieval results in our hybrid framework as future work.

---

### Author Response · Authors · 2020-11-24
**Common Response-Evaluation on existing benchmarks and Ablation study on Retrieval-based Augmentation**

We thank all reviewers for the insightful and valuable comments!

Q1 (R1, R3, R4): Evaluation on existing benchmarks.

A1: We conducted additional experiments on a public dataset, i.e., the Python Code Summarization Dataset (PCSD), which was also used in Rencos (the most competitive baseline in our paper). The total number of code samples in PCSD is 109,726[1]. This number is comparable to the size (i.e., ~95k) of our own CCSD benchmark.

Setting: We follow the setting of Rencos and split PCSD into the training set, validation set and testing set with fractions of 60%, 20% and 20%. We construct the static graph and compare our methods on PCSD against various competitive baselines, i.e., NNGen, CodeNN, Rencos and Transformer, which are either retrieval-based, generation-based or hybrid methods.

Results: The results are shown as below. We can see that our method outperforms NNGen, CODENN, Rencos and Transformer by 0.95, 3.27 and 1.12 in terms of BLEU-4, ROUGE-L and METEOR. We also perform the ablation study on PCSD to demonstrate the usefulness of the static graph (i.e., HGNN w/o dynamic) and dynamic graph (i.e., HGNN w/o static). The results also demonstrate that both the static graph and the dynamic graph can contribute to our framework. In summary, the results on both our released benchmark (C benchmark) and existing benchmark (PCSD) demonstrate the effectiveness of our method.

Methods \ Metrics | BLEU-4 | ROUGE-L | METEOR

NNGen | 21.60 | 31.61 | 15.96

CODE-NN| 16.39 | 28.99 | 13.68

Transformer| 17.06 | 31.16 | 14.37

Rencos | 22.24 | 36.00 | 18.26

HGNN w/o static | 21.82 | 38.61 | 18.36

HGNN w/o dynamic | 21.75 | 38.37 | 18.42

HGNN | 23.19 | 39.27 | 19.38


Q2 (R1, R4): Ablation study on Retrieval-based Augmentation.

A2: We follow the suggestion and add the experiments to evaluate the impact of the code-based augmentation and summary-based augmentation on our CCSD dataset. We show the results in the in-domain dataset, out-of-domain dataset and the mix of in-domain and out-of-domain dataset as below.

Overall, we found that: retrieval-augmented mechanism significantly contributed to the overall model performance (HGNN vs. HGNN w/o augment). More specifically, we noticed that summary-based augmentation has the most impact (HGNN vs. HGNN w/o summary augment). Besides, considering both summary and code augmentation further significantly improved the performance compared to considering only summary augmentation (HGNN vs. HGNN w/o code augment). The summary-based augmentation is more useful, we conjecture that it depends on the specific task: 1) this task is to generate summary and 2)  the code and summary are heterogeneous data. Thus, summary-based augmentation could provide a more direct signal for generating better summaries. However, the code-based augmentation could further improve the performance by enhancing the semantic learning of the program. Combining them together, our method achieves the best result.


In domain:

Methods \ Metrics | BLEU-4 | ROUGE-L | METEOR

HGNN w/o augment | 12.43 | 30.05 | 25.75

HGNN w/o summary augment | 13.37 | 30.36 | 26.13

HGNN w/o code augment | 15.10 | 32.19 | 27.83

HGNN | 16.24 | 33.62 | 29.60

Out of  domain:

Methods \ Metrics | BLEU-4 | ROUGE-L | METEOR

HGNN w/o augment | 5.56 | 22.64 | 18.27

HGNN w/o summary augment | 5.81 | 22.97 | 19.05

HGNN w/o code augment | 6.94 | 23.80 | 20.44

HGNN | 7.62 | 24.77 | 20.78

Overall:

Methods \ Metrics | BLEU-4 | ROUGE-L | METEOR

HGNN w/o augment | 9.87 | 27.04 | 23.16

HGNN w/o summary augment | 10.34 | 27.43 | 23.82

HGNN w/o code augment | 12.01 | 28.79 | 24.93

HGNN | 13.39 | 30.23 | 26.22


[1]. A Parallel Corpus of Python Functions and Documentation Strings for Automated Code Documentation and Code Generation. Barone et al. IJCNLP(2) 2017.

---

### Author Response · Authors · 2020-11-24
**Common Response-Summary of Updated Version**

Thank all reviewers for your suggestions and comments. We have submitted an updated version of the paper and included additional experiments.

1 ) We have added an evaluation on existing benchmarks according to Reviewer 1’s, Reviewer 3’s and Reviewer 4’s comments.

2 ) We have added an ablation study on Retrieval-based Augmentation according to Reviewer 1’s and Reviewer 4’s comments.

3 ) We have provided the mathematical formulation of “Step 3: Retrieved Summary-based Augmentation” according to Reviewer 1’s comments.

---

### Author Response · Authors · 2021-01-15
**Recomputing the METEOR scores on the CCSD benchmark using the official Meteor 1.5 script (previously the NLTK version was used)**

Dear Program Chairs and reviewers,

We are writing to report that we realized
**there was a discrepancy between the METEOR scores** computed by the NLTK package (which we used to compute METEOR scores for all baselines and our methods on the CCSD benchmark) and the METEOR scores computed by the official Meteor 1.5 script. The reason is that the NLTK version follows the original METEOR paper while the official script follows the Meteor 1.3 paper which made some changes to improve the original version.

Even though **the two versions of METEOR are both valid and effective**, to follow most previous natural language generation papers, we decided to recompute the METEOR scores for all methods on the CCSD benchmark using the official Meteor 1.5 script.

We have done the following things to address this issue:
We have recomputed the METEOR scores using the official Meteor 1.5 script for **all baselines and our methods on the CCSD benchmark**, and updated the corresponding numbers in the manuscript. Please be informed that **this change did NOT make any difference to the experimental conclusion** because in our experiments, we computed the METEOR scores for all baselines and our methods with the same script, even though the absolute values of their METEOR scores changed with the new evaluation script, the relative relations of these values did not change. And our model still achieved the state-of-the-art performance on the CCSD benchmark. For example, on the CCSD benchmark, our proposed model still outperformed the existing state-of-the-art method by **1.24** in terms of the updated METEOR scores.
We have double-checked the results of all the other evaluation metrics carefully to ensure their correctness.


References:
- NLTK version of METEOR: https://www.nltk.org/_modules/nltk/translate/meteor_score.html
- Official Meteor 1.5 script: https://www.cs.cmu.edu/~alavie/METEOR/README.html
- Original METEOR paper: http://www.cs.cmu.edu/~alavie/METEOR/pdf/Lavie-Agarwal-2007-METEOR.pdf
- Meteor 1.3 paper: http://www.cs.cmu.edu/~alavie/METEOR/pdf/meteor-wmt11.pdf

---

### Decision · Program_Chairs · 2021-01-07
**Final Decision**

**Decision:**

Accept (Spotlight)

**Comment:**

This paper proposes an interesting method for combining retrieval-based models and graph neural networks for source code summarization. Finding new ways of bringing in additional context for graph-based models is an important research direction in this space, and the paper presents a novel and effective approach. The initial submission was missing experiments on existing benchmarks, but new experiments presented in the discussion phase are enough to resolve that concern. Reviewers are unanimously in support of acceptance.